# Cell Biological and Antibacterial Evaluation of a New Approach to Zirconia Implant Surfaces Modified with MTA

**DOI:** 10.3390/biomimetics9030155

**Published:** 2024-03-01

**Authors:** Beatriz Ferreira Fernandes, Neusa Silva, Mariana Brito Da Cruz, Gonçalo Garret, Óscar Carvalho, Filipe Silva, António Mata, Helena Francisco, Joana Faria Marques

**Affiliations:** 1Oral Biology and Biochemistry Research Group—Unidade de Investigação em Ciências Orais e Biomédicas (UICOB), Faculdade de Medicina Dentária, Universidade de Lisboa, 1600-277 Lisboa, Portugal; 2Department of Mechanical Engineering, Center for Microelectromechanical Systems (CMEMS), University of Minho, 4800-058 Guimarães, Portugal; 3Oral Biology and Biochemistry Research Group—Unidade de Investigação em Ciências Orais e Biomédicas (UICOB), LIBPhys-FCT UIDB/04559/2020, Faculdade de Medicina Dentária, Universidade de Lisboa, 1600-277 Lisboa, Portugal; 4CEMDBE—Cochrane Portugal, Faculdade de Medicina Dentária, Universidade de Lisboa, 1600-277 Lisboa, Portugal; 5Grupo de Investigação Implantologia e Regeneração Óssea (UICOB), Faculdade de Medicina Dentária, Universidade de Lisboa, 1600-277 Lisboa, Portugal

**Keywords:** mineral trioxide aggregate, dental implant, zirconia, osteoblasts, fibroblasts

## Abstract

Peri-implantitis continues to be one of the major reasons for implant failure. We propose a new approach to the incorporation of MTA into zirconia implant surfaces with Nd:YAG laser and investigate the biological and the microbiological responses of peri-implant cells. Discs of zirconia stabilized with yttria and titanium were produced according to the following four study groups: Nd:YAG laser-textured zirconia coated with MTA (Zr MTA), Nd:YAG laser-textured zirconia (Zr textured), polished zirconia discs, and polished titanium discs (Zr and Ti). Surface roughness was evaluated by contact profilometry. Human osteoblasts (hFOB), gingival fibroblasts (HGF hTERT) and *S. oralis* were cultured on discs. Cell adhesion and morphology, cell differentiation markers and bacterial growth were evaluated. Zr textured roughness was significantly higher than all other groups. SEM images reveal cellular adhesion at 1 day in all samples in both cell lines. Osteoblasts viability was lower in the Zr MTA group, unlike fibroblasts viability, which was shown to be higher in the Zr MTA group compared with the Zr textured group at 3 and 7 days. Osteocalcin and IL-8 secretion by osteoblasts were higher in Zr MTA. The Zr textured group showed higher IL-8 values released by fibroblasts. No differences in *S. oralis* CFUs were observed between groups. The present study suggests that zirconia implant surfaces coated with MTA induced fibroblast proliferation and osteoblast differentiation; however, they did not present antibacterial properties.

## 1. Introduction

Bacterial colonization is one of the major reasons for implant loss, and it is a necessary condition for the development of peri-implantitis. Peri-implantitis is an inflammatory process that affects the tissues surrounding an osseointegrated implant, resulting in loss of supporting bone [1,2]. Thus, the biological seal surrounding the implant is considered an important factor for the long-term success of peri-implant health [3].

The literature suggests that the epithelial barrier formed by the peri-implant tissues is less stable compared with the junctional epithelium [3]. This limitation can be explained by the attachment of the non-keratinized peri-implant epithelium to the implant surface. This adhesion occurs via internal basal lamina and hemidesmosomes and is limited to the most apical region. The orientation of the collagen fibers of peri-implant connective tissue, mostly circular or oblique, also contributes to the formation of a weak biological seal around the implant when compared with Sharpey’s fibers, which are orientated perpendicularly to the tooth surface [3].

Titanium alloys are the most common biomaterials for dental implants due to their mechanical properties and excellent biocompatibility [4,5]. However, the gray color of the material and hypersensitivity reactions caused by the release of ions are a concern [4]. Furthermore, the literature reports that corrosion of titanium implants can lead to changes in the oral microbiome. Implant base material alternatives that simultaneously combine mechanical and excellent biological properties have been investigated [4]. Zirconia has proven to be a possible alternative to titanium due to its mechanical properties, because its osseointegration ability is similar to titanium, and because of its good biocompatibility and superior soft tissue response and aesthetic characteristics [4,5,6] These characteristics of zirconia could be an advantage in terms of rehabilitation in the esthetic zone and in cases where patients are sensitive to metal [5]. Additionally, zirconia has a low surface energy, which results in reduced bacterial colonization [6]. Several in vitro studies comparing zirconia and titanium have demonstrated that zirconia surfaces can lead to a significant decrease in the adhesion of periodontal pathogens [5]. 

Despite all of the beneficial properties previously described, zirconia suffers from a degradation process at low temperatures—what one might term an aging process. The degradation process is due to a slow transformation from a tetragonal (stable between 1170 °C–2370 °C) to a monoclinic (stable at temperatures < 1170 °C) phase. This phenomenon results in the formation of cracks and eventual fracture of the ceramic [4]. Zirconia is also a biologically inert material [7]. Thus, several surface treatment techniques and protocols have been developed to increase its bioactivity. Sandblasting, acid etching, laser treatment and surface coatings are strategies used to improve the surface bioactivity of titanium and zirconia implants [6,7]. Implant base material and surface properties can significantly influence the success of osseointegration [4,7]. A recent review suggests that the results obtained at the tissue–implant interface are better or similar on treated zirconia surfaces, when compared with titanium surfaces given the same treatment [4]. 

Several strategies involving mechanisms of repulsion or elimination of bacteria have been developed to optimize the biological response and to prevent bacterial adhesion and biofilm formation on implant and abutment surfaces [8]. The first is based on the prevention of biofilm formation, while the second consists in the development of antibacterial surfaces that lead to the disruption of bacterial cells [8]. Surface functionalization with coatings that provide antibacterial properties is a potential technique for eliminating bacteria adhesion to implants and surrounding areas [8,9]. The coating acts as a reservoir of bactericidal agents and allows their local and sustained release, contributing to a more efficient action [8]. There is no defined optimal strategy by which to prepare antibacterial surfaces and, for this reason, it is essential to develop and investigate different approaches, both isolated and in combination, while fully characterizing these new surfaces [8].

Mineral trioxide aggregate (MTA) is composed of tricalcium silicate, dicalcium silicate, tricalcium aluminum, calcium oxide, silicon dioxide, aluminum trioxide, and a radiopacifying agent [10,11,12,13]. Due to its excellent chemical, physical and biological properties, in particular its biocompatibility and osteoconductive potential, it has been used in dentistry for over 20 years [10,11,14,15,16,17,18]. MTA is used in endodontic treatments [13,19,20] and is also recommended in vital pulp therapy to preserve the vitality of dental pulp [13,19,20]. Several studies have demonstrated the antibacterial and antifungal properties of this material [19,20,21,22]. Despite the potential of MTA as an antibacterial agent, there are no studies on its use in implantology [23]. 

In this study, we propose a new surface treatment approach that combines two techniques: texturing using Nd:YAG laser and the use of MTA as a bioactive agent [24]. The use of laser to create the texture is based on its ability to produce a controlled texture without direct contact with the surface, as shown in previous studies [25]. On the other hand, the surface texture is expected to micro-mechanically retain the MTA coating. In this sense, this study investigates the biological response of peri-implant cells in contact with MTA-coated zirconia surfaces, as well as their microbiological adhesion response.

## 2. Materials and Methods

### 2.1. Samples Processing

Zirconia discs were produced using the cold-pressing technique from a commercial 3 mol% Y_2_O_3_-stabilized zirconia spray-dried powder (TZ-3YB-E, Tosoh Corporation©, Tokyo, Japan). Table 1 presents the chemical composition of zirconia powder used in this study. Discs with 3 mm of thickness and 8 mm diameter were compacted in a stainless-steel mold using a uniaxial pressure of 25 MPa for 1 min.

Before sintering, green test samples were laser textured using a Nd:YAG laser (OEM Plus, Sisma, Vicenza, Italy), with a focus distance of 328 mm, a 3 μm focal spot size, 6 W of output power, a pulse width of around 35 ns and a wavelength of 1064 nm. Surface texture consisted of a square crosslinked pattern with 16 lines in each direction over the surface (Figure 1). The texturing was performed in normal air and atmospheric pressure using the following laser parameters: power of 40% (2.4 W), a scan speed of 128 mm/s and three laser passages, based on previous works [26,27,28]. 

Samples were then sintered in a high-temperature furnace (Zirkonofen 700, Zirkonzahn^®^, Gais, Italy) in the air at 1500 °C for 2 h with a heating and cooling rate of 8 °C/min. After sintering, the surfaces were ultrasonically cleaned for 1 min. 

Mineral trioxide aggregate powder (MTA Angelus^®^, Angelus, Londrina, Brazil) was used to produce the bioactive coating in laser-textured surfaces. The atomic composition of MTA powder used in this study is described in Table 2. 

MTA powder was prepared according to the manufacturer’s instructions and quantities. Hence, 1 spoon of the power was placed in 1 drop of distilled water and mixed for about 30 s. The MTA creamy mixture was applied to the textured surface using a spatula. To enhance the adherence of the coating to the surface, we developed a homemade device composed of parallel bars attached to the metallic support through springs (Figure 2).

The samples were placed inside small blue tube sections (obtained from PCR tube strips), and a punch was positioned on the uncoated side of the zirconia substrate (textured surface). The strips were then placed under the intermediate screws of the developed device, and pressure was applied for 3–4 h, through the pressure of the plates. After this time, the pressure was released and the MTA-coated zirconia structured samples were kept inside the tubes for 24 h to ensure effective setting of the coating. Finally, the samples were removed and subjected to a highly energetic and aggressive ultrasonic adhesion test for 1 min.

Titanium discs with 2 mm of thickness and 8 mm of diameter were produced from Ti grade V (Ti6Al4V) powder using the hot-pressing technique, according to previously described methods [9,29]. The mold containing powder was heated to 1200 °C at 31 °C/min. At 1100 °C, the pressure on the samples was raised to 20 MPa for 30 min. Titanium discs were wet ground on SiC papers to 400 mesh and then polished to a near-mirror finish using aluminum oxide suspension (1 μm). Samples were then cleaned ultrasonically. 

In this study, four sample groups were considered for comparison purposes of the experimental results, as shown in Table 3. 

### 2.2. Samples Characterization

#### Surface Roughness

Surface roughness was evaluated using the Ra parameter—the arithmetic mean value between the peak and valley height values in the effective roughness profile. The Ra roughness was measured in accordance with ISO 4288:1996 standard using a mechanical 2D profilometer (Surftest SJ 201, Mitutoyo, Kanagawa, Japan). The measurements were recorded in different regions, always changing the scanning directions. The scanning speed used was 0.25 mm/s and 3 × 1.5 mm lines were scanned.

### 2.3. Cell Cultures

Human fetal osteoblasts (hFOB 1.19—ATCC^®^, CRL-11372TM; American Culture Collection, Manassas, VA, USA) were cultured in 75 cm^2^ culture flask (VWR^TM^, Radnor, PA, USA) in an atmosphere of 5% of CO_2_, 98% of humidity and temperature of 37 °C with a culture medium composed of a mixture (1:1 *v*/*v*) of Ham’s F12 Medium (Sigma-Aldrich^®^, St. Louis, MO, USA) and Dulbecco’s Modified Eagle’s Medium (DMEM—BioWhittaker^®^, Lonza^TM^, Basel, Switzerland) supplemented with 10% of bovine fetal serum (Biowest©, Nuaillé, France) and 0.3 mg/mL of G418 (InvivoGen, Toulouse, France).

Immortalized human gingival fibroblasts (HGF hTERT—T0026; Applied Biological Materials Inc., Richmond, BC, Canada) were cultured under atmospheric conditions as previously described for osteoblasts in a culture medium composed of Dulbecco’s Modified Eagle’s Medium (DMEM—BioWhittaker^®^, Lonza^TM^, Basel, Switzerland) supplemented with 10% of bovine fetal serum (Biowest©, France) and 100 UmL^−1^ penicillin, and 100 μg/mL streptomycin (Lonza^TM^, Basel, Switzerland).

When cells reached 80% of confluence, cells were detached using trypsin-EDTA (Lonza^TM^, Basel, Switzerland), centrifuged approximately 100× *g* for 5 min and re-suspended in culture media. Cells were seeded on discs and distributed in 48-well culture plates (Corning^®^, Corning, NY, USA) at a density of 1 × 10^4^ cells/well for biological assays. The experiments were conducted using a fifth passage. Cells cultured directly on the treated polystyrene surface of the well were used to form a positive control.

#### 2.3.1. Cell Viability

Cell viability was evaluated in three independent experiments with duplicates of 5 samples per group (n = 15). CellTiter-Blue^®^ reagent (Promega, Madison, WI, USA)—a viability assay based on the conversion of resazurin into a fluorescent product—was used according to the supplier’s instructions. The conversion rate after 1, 3, 7 and 14 days of culture was quantified as fluorescence intensity in arbitrary fluorescence units (AU). Fluorescence intensity was detected at an excitation wavelength of 530/30 nm and an emission wavelength of 595/10 nm using a multimode microplate reader (VICTOR Nivo^TM^ HH3500, PerkinElmer^®^, Beaconsfield, UK). 

#### 2.3.2. Cell Morphology

Osteoblasts and fibroblasts were cultured on discs for 1 day. Culture wells were washed with phosphate buffered saline (PBS—VWR^®^, Radnor, PA, USA) and then fixed with 2.5% glutaraldehyde (VWR^®^, USA) for 1 h. A dehydration process took place via serial dilution of ethanol. Samples were metallized using a gold target in a JEOL JFC 1200 (Jeol Ltd., Tokyo, Japan) sputtering chamber. Samples were observed under JEOL JSM5200-LV (Jeol Ltd., Tokyo, Japan) and secondary images were carried out at an acceleration voltage of 15 kV and 25 kV and different magnifications (100, 150, 180, 200×). Two calibrated researchers performed the image analysis considering cell morphology and adhesion to the materials and cell spreading.

#### 2.3.3. Interleukin 8

Interleukin 8 (IL-8) is an important neutrophil chemotactic factor that responds to inflammation. The quantification of IL-8 was achieved at 1 and 3 days of osteoblast and fibroblast culture by human IL-8/CXCL8 DuoSet ELISA (R&D Systems, Inc., Minneapolis, MN, USA), using a multimode microplate reader (VICTOR Nivo^TM^ HH3500, PerkinElmer^®^, Beaconsfield, UK). The results were obtained in absorbance units (AU), relative to the values of light intensity, and were converted in pg/mL according to the calibration curve performed. One experiment with duplicates of 4 samples per group was performed (n = 4). 

#### 2.3.4. Osteocalcin

Osteocalcin is a non-collagenous protein in bone and is expressed in osteoblasts. Osteocalcin quantification was carried out in osteoblasts culture after 1 and 3 days, using Human Osteocalcin DuoSet ELISA (R&D Systems, Inc., Minneapolis, MN, USA), by a technique of luminescence. The results were obtained in absorbance units (AU) using a multimode microplate reader (VICTOR Nivo^TM^ HH3500, PerkinElmer^®^, UK) and converted in pg/mL according to the curve of calibration performed. One experiment with duplicates of 4 samples per group was performed (n = 4). 

### 2.4. Bacterial Strain and Growth Conditions

To carry out this investigation, the *Streptococcus oralis* CECT 907T strain was cultured on a plate with an enriched blood agar in an anaerobic atmosphere at 37 °C for 72 h, with a gas mixture of 10% CO_2_, 10% H_2_, and the remainder N_2_. Afterwards, a single colony was transferred to 15 mL of brain–heart infusion modified medium (BHI-2) in anaerobic conditions at 37 °C until it reached the exponential growth phase. The optical density (OD) of the suspension was measured at 550 nm using a Camspec M50 spectrophotometer to confirm the growth, and the OD was standardized to 0.4. 

#### Colony Forming Unit (CFU)

In this experiment, sample discs from all of the groups were randomly selected and placed in a 24-well plate (Corning^®^, Corning, NY, USA) with *S. oralis* in the exponential phase and incubated in anaerobic conditions at 37 °C. The efficacy of MTA as a bactericidal or bacteriostatic agent was assessed by counting the colony forming units (CFUs) after 24 h of *S. oralis* culture. To determine the number of viable bacterial cells that adhered to the discs over time, the discs were washed once with filtered PBS (VWR^®^, Radnor, PA, USA) and placed in a falcon with 3 mL of PBS (VWR^®^, Radnor, PA, USA). The falcon was then vortexed at 16 rpm × 100 for 1 min, followed by ultrasonication for 4 min and another vertexing for 2 min at 16 rpm × 100. Ten-fold serial dilutions were made up to 10^−6^, and 20 μL of each dilution were plated in triplicate on supplemented brain–heart infusion agar (BHIA), which were then incubated at 37 °C under anaerobic conditions.

### 2.5. Statistical Analysis

Statistical analysis was conducted using IBM^®^ SPSS^®^ 27.0 statistics software for Mac (SPSS, Chicago, IL, USA). The Kolmogorov–Smirnov test was used to test data for normality. Comparisons between groups for roughness values, cell viability, interleukin 8, osteocalcin levels and CFUs were carried out using a factorial analysis of variance ANOVA or Kruskal–Wallis tests as appropriate, and significant differences between groups were identified with Tukey’s post-hoc test. The significance level was set as *p* < 0.05. All data are presented as mean ± standard deviation (SD).

## 3. Results

### 3.1. Samples Characterization

#### Surface Roughness

Before biological assays, surface roughness measurement was performed for all samples. Roughness values of Ra (μm) presented as mean and standard deviation (SD) are described in Table 4. The results show similar Ra values between Zr MTA, Zr and Ti samples, without any significant differences between them (*p* > 0.05). However, Zr textured Ra values were significantly higher compared with the other groups (*p* < 0.05).

### 3.2. Cell Culture

#### 3.2.1. Cell Viability

Cell viability results were obtained for 1, 3, 7 and 14 days for osteoblasts culture and 1, 3 and 7 days for fibroblasts culture (as shown in Figure 3A and Figure 3B, respectively). Cell viability decreased from 1 to 3 days in both cultures and then increased over time for all groups. On osteoblasts culture, Zr MTA samples showed significantly lower viability compared with the Ti group (*p* < 0.05) at 1 day; Zr (*p* < 0.05) and Ti group (*p* < 0.001) at 3 days; and Zr textured (*p* < 0.001), Zr (*p* < 0.005) and Ti (*p* < 0.001) groups at 7 days and 14 days, respectively. These results contrast with those of the Ti group, which showed significantly higher viability values compared with Zr MTA in all measured time points and when compared with Zr textured (*p* < 0.001) and Zr (*p* < 0.01) at 3 days and with Zr (*p* < 0.05) at 14 days. Regarding fibroblast culture, Zr MTA showed significantly higher values compared with Zr textured at 3 and 7 days (*p* < 0.05). 

#### 3.2.2. Cell Morphology

SEM images (Figure 4) obtained on samples after 1 day of osteoblast and fibroblast cultures are presented with their respective magnifications. Images show adherent cells in all samples after 1 day of culture in both osteoblast and fibroblast cultures. However, Zr MTA samples appear to have fewer osteoblasts compared with the other surfaces, unlike fibroblast cell bodies, which appear to have homogeneously adhered to all of the samples. Concerning cell morphology, the images show that osteoblast culture in Zr and Ti samples presented an elongated shape with some projecting processes, while in Zr MTA and Zr textured a more prismatic cell conformation was observed in the osteoblasts. Fibroblasts presented a typical elongated veil shape and filopodia formation, with more evident cell bodies in the Zr MTA and Zr textured samples.

#### 3.2.3. Interleukin 8

IL-8 secretion by osteoblasts and fibroblasts was obtained at 1 and 3 days and is presented in Figure 5 as an IL-8 concentration/cell viability ratio. The results reveal that the Zr MTA group showed higher IL-8 secretion by osteoblasts compared with the Ti group (*p* < 0.05) at 1 day. At 3 days, the Ti group revealed significantly lower values compared with the Zr MTA (*p* < 0.001), Zr textured (*p* < 0.001) and Zr (*p* < 0.005) groups. Concerning fibroblast IL-8 secretion, the Zr textured group showed significantly higher values compared with Zr (*p* < 0.05) and Ti (*p* < 0.001) at 1 day and compared with all groups at 3 days of culture. 

#### 3.2.4. Osteocalcin

Osteocalcin results are also presented as osteocalcin concentration per cell viability (Figure 6). The results show an increased production of osteocalcin from day 1 to 3 in all groups. Although no significant differences between groups were found at 1 day of culture, higher levels of osteocalcin were found in the Zr MTA group compared with the Zr (*p* < 0.05) and Ti (*p* < 0.001) groups at 3 days. 

### 3.3. Bacterial Growth

#### Colony Forming Unit (CFU)

The CFUs per milliliter of *Streptococcus oralis* on the discs were evaluated at 24 h of culture, as shown in Figure 7. All samples showed CFU values of around 10^6^, and no statistically significant differences were observed between groups.

## 4. Discussion

Despite continuous research on implant biomaterial improvements, bacterial colonization continues to be one the most relevant causes of dental implant failure [4]. Zirconia dental implant surfaces have demonstrated low bacterial colonization along with their biological, physical, and aesthetic properties [6] However, zirconia requires surface treatments to increase its bioactivity [7]. Surface functionalization with coatings has been proposed to be a possible approach for improving peri-implant cell response and, simultaneously, for reducing bacterial adhesion and biofilm formation [8,9].

MTA has shown biocompatibility and antibacterial properties, and has been extensively used in dentistry for more than two decades [13,19]. Based on these properties, we proposed the use of MTA as an antibacterial coating and functionalization approach for implant and abutment zirconia surfaces. In the present study, yttria-stabilized zirconia samples were laser-textured and coated with MTA and tested in biological assays with the aim of better understanding the cellular behavior of peri-implant tissues in contact with these implant surfaces. 

To mimic the in vivo peri implant cell environment, immortalized human fetal osteoblasts (hFOB 1.19) and immortalized human gingival fibroblasts (HGF hTERT) were used. Our results show that osteoblast viability was significantly lower in the Zr MTA group in all of the measured timepoints. These results are in line with SEM micrographs, which reveal that a smaller number of osteoblasts adhered to the Zr MTA samples after 24 h of culture compared with other groups. These results are, however, contrary to those found in other studies [12,20,30]. Besides the use of different cell lines and the use of a fresh or set material, previously published studies have evaluated the cellular response in direct contact with pure MTA cements, instead of MTA-based coatings applied on a textured zirconia disc, as in the specific context of this study. Additionally, many studies have evaluated the cellular response to MTA using eluate assays at different dilutions. Higher concentrations of MTA have been observed to release a higher level of toxic substances leading to lower cell viability [13]. Therefore, the comparison of our results with previous literature is limited, as the experimental designs have several differences from our study. Another explanation for the contradictory results could rely on the pH of MTA. The high alkaline pH could explain the cytotoxic effect of MTA and a reduction in cell viability in direct contact assays [13,19]. Although osteoblast viability was lower in the Zr MTA group, fibroblast viability was shown to be significantly higher in the Zr MTA group compared with the Zr textured group at 3 and 7 days. Khedmat et al. and Youssef et al. evaluated the cell viability of gingival fibroblasts in contact with MTA Angelus and obtained similar results [13,31]. The difference in cell viability between osteoblasts and fibroblasts can be explained by the following hypotheses: (1) the phagocytic activity of osteoblasts may inhibit cell proliferation because cells keep their metabolic energy for phagocytosis and intracellular digestion [32], (2) the presence of MTA accelerates the process of differentiation of osteoblasts and consequently reduces their proliferation [32,33], (3) fibroblasts have a greater speed of adhesion growth and proliferation compared with osteoblasts [25], and (4) the difference is due to surface roughness—Zr MTA discs are smooth surfaces (roughness of 0.31μm). The studies in the literature report that bone cells show better behavior with increased roughness, while fibroblasts demonstrate a good biological response associated with smooth surfaces [4].

Despite the fact that osteoblast viability results were not found to be promising in the Zr MTA group, osteocalcin release was significantly higher in this group compared with the other groups at 3 days. Significantly higher values of IL-8 release by osteoblasts were also observed in comparison with the Ti group in the first day of culture. The existing literature [34] demonstrates that osteoblasts in contact with MTA are stimulated to release greater amounts of cytokines (involved in bone turnover) as well as osteocalcin, which plays a regulatory role in bone mineralization [35]. These observations suggest that, although there are fewer osteoblasts in the Zr MTA group, these osteoblasts are more differentiated, supporting this hypothesis. The IL-8 secretion in fibroblast culture indicates that the Zr textured group showed significantly higher values compared with Zr and Ti groups at 1 day and compared with all groups at 3 days of culture. These results suggest that MTA does not induce differentiation in fibroblasts, but that the texture associated with the increased surface roughness may help in this process. 

To evaluate the antibacterial effect of MTA coating, we evaluated the adhesion and growth of viable *S. oralis* on the samples. The choice of this bacterial strain is based on its role as a primary colonizer in the oral cavity.

No differences in CFUs of *S. oralis* culture were observed between groups, suggesting that this MTA coating technique does not seem to confer antibacterial properties to the samples. A study by Hasna et al. demonstrated a significant reduction in bacterial viability when in contact with MTA cement. However, their assessment was carried out in *P. gingivalis*, *P. endodontalis*, *P. micra*, *F. nucleatum* and *P. intermedia* biofilms using the MTT test [19]. In vitro studies in the literature report that zirconia surfaces can lead to a significant decrease in periodontal microbiome adhesion when compared with titanium surfaces [5]. In our study, we were not able to obtain these results—no differences were observed between zirconia and titanium surfaces (Figure 7). However, the evaluation of bacterial growth was only carried out at a single time and the evaluation of biofilm mass should be considered in future works. 

This study evaluates, for the first time, the effect of MTA-coated textured zirconia surfaces in the cell responses of soft and hard human peri-implant tissues. The results of this study demonstrate that the proposed MTA coating technique in zirconia implant surfaces appeared to induce differentiation in osteoblasts, but that the same effect was not observed in fibroblasts, despite inducing their proliferation. This MTA coating technique did not appear to confer antibacterial properties against *S. oralis*. It should be noted that this was an in vitro study and that therefore the results must be interpreted within the limitations of this study design. Another limitation refers to the coating stability of this strategy, which was not optimal and that could have impacted the results. A larger sample size, as well as other differentiation and inflammatory markers should be included in future works. In vivo cell behavior integrated into complex biological systems should also be evaluated to validate these findings. In this sense, further studies with new approaches to the incorporation of MTA onto implant surfaces, improved mechanical and surface characterization methods and evaluation of cellular response should be carried out. 

## 5. Conclusions

The results obtained in this study demonstrate that, despite the potential beneficial properties of MTA regarding biocompatibility and antibacterial capacity, the proposed strategy of coating MTA onto textured zirconia implant surfaces failed to demonstrate additional antibacterial properties to the samples. However, the addition of MTA to zirconia laser textured samples increased differentiation in osteoblasts and proliferation in gingival fibroblasts. 

## Figures and Tables

**Figure 1 biomimetics-09-00155-f001:**
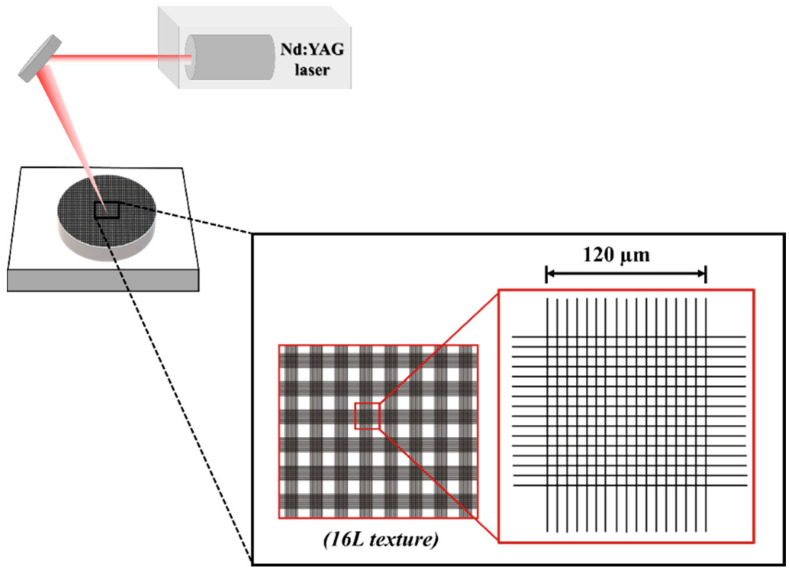
Schematic representation of the laser surface texturing process and the correspondent design of texture.

**Figure 2 biomimetics-09-00155-f002:**
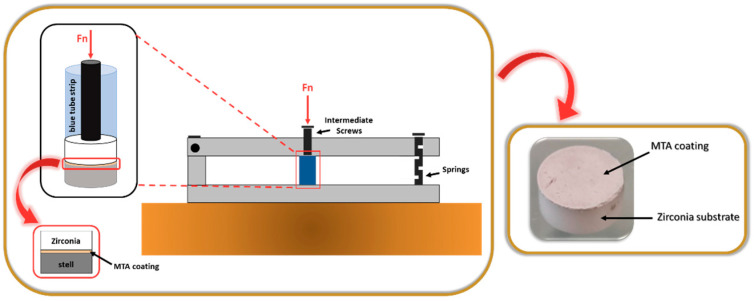
Homemade device used to perform the pressing of the MTA coating.

**Figure 3 biomimetics-09-00155-f003:**
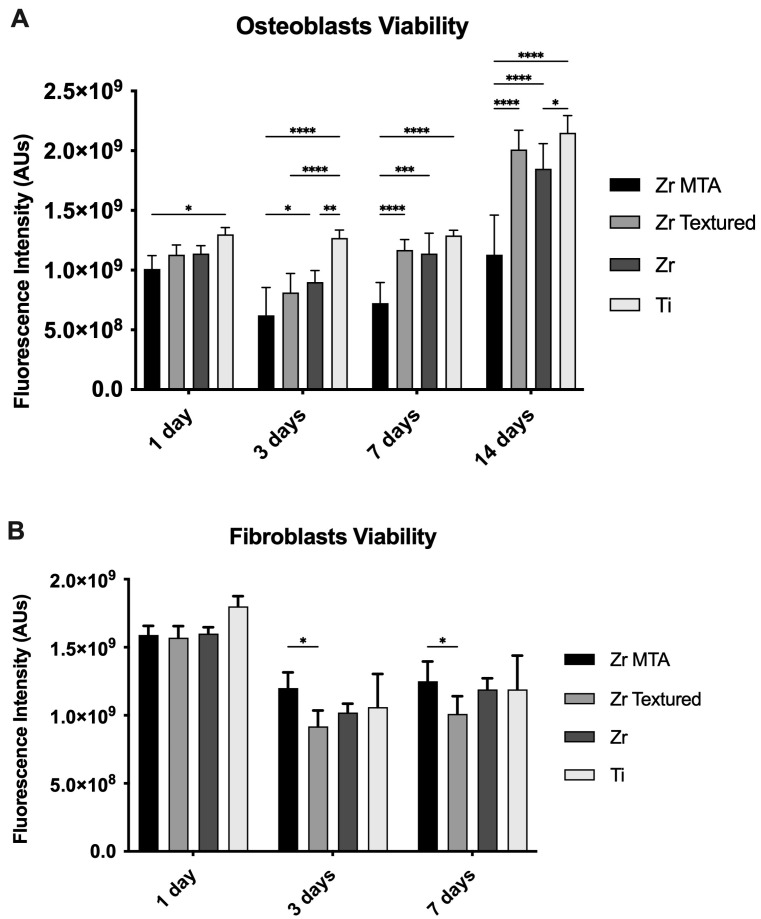
Bar charts showing osteoblast (**A**) and fibroblast (**B**) viability measured on Zr MTA, Zr textured, Zr and Ti as mean exhibited in arbitrary units (AU) of fluorescence intensity. The standard deviation (SD) is represented by error bars. Statistical significance: * *p* < 0.05, ** *p* < 0.01, *** *p* < 0.0005, **** *p* < 0.0001; n = 15.

**Figure 4 biomimetics-09-00155-f004:**
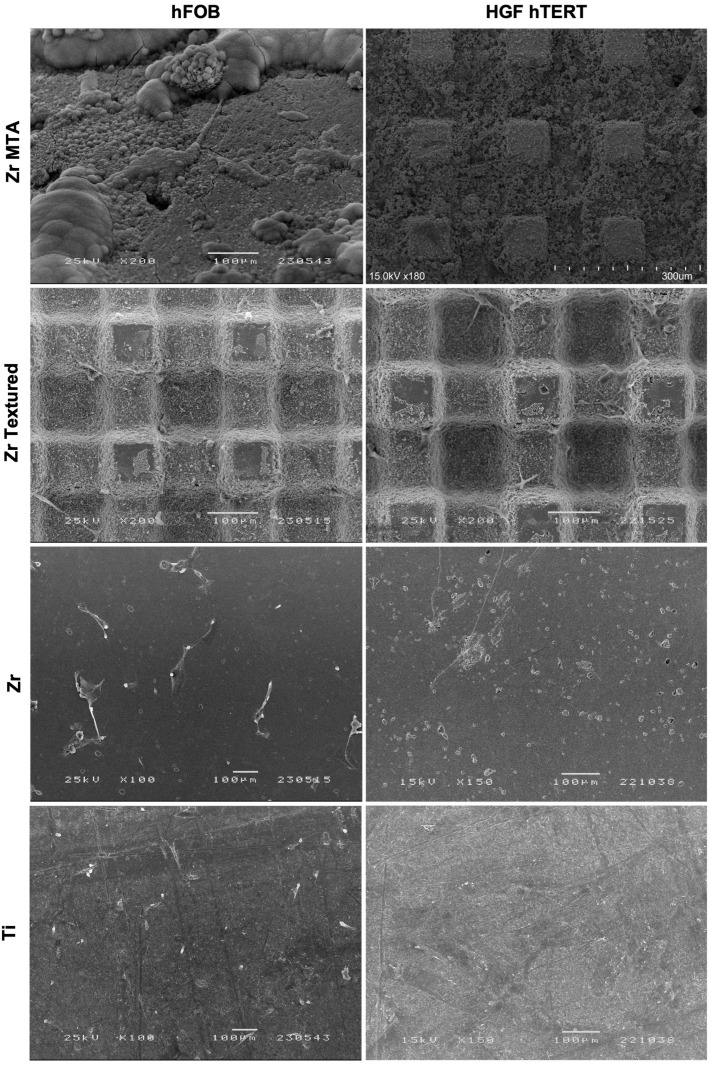
SEM micrographs with osteoblasts (hFOB) and fibroblasts (HGF hTERT) cultured on surfaces at 1 day (100×, 150×, 180× and 200× magnification).

**Figure 5 biomimetics-09-00155-f005:**
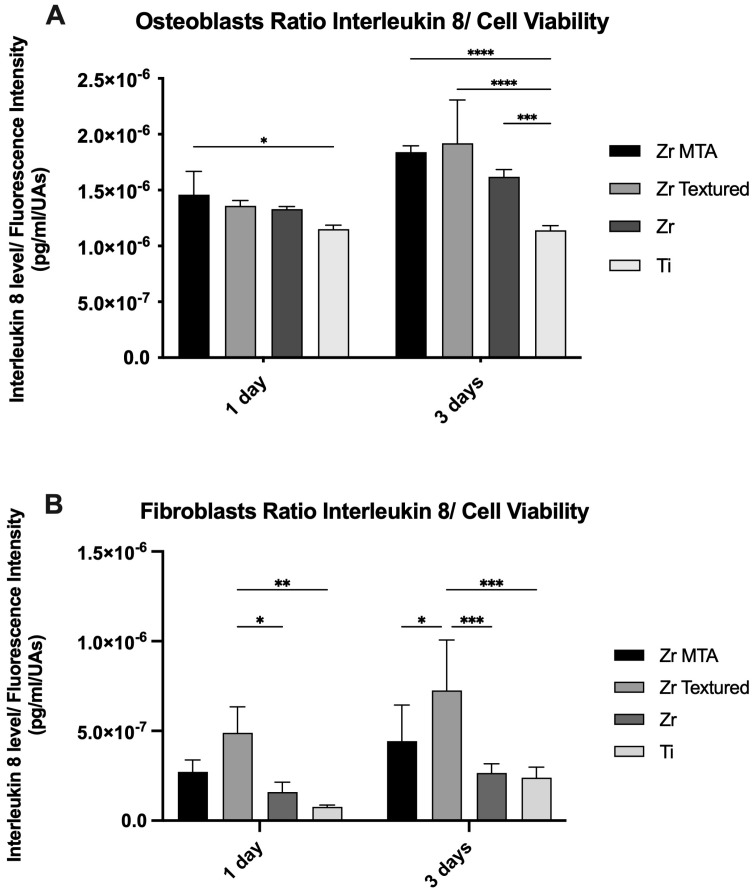
Bar charts showing interleukin 8 secretion by osteoblasts (**A**) and fibroblasts (**B**) as a mean concentration in pg/mL/UAs. Error bars represent standard deviation (SD). Statistical significance: * *p* < 0.05, ** *p* < 0.001, *** *p* < 0.005, **** *p* < 0.001; n = 4.

**Figure 6 biomimetics-09-00155-f006:**
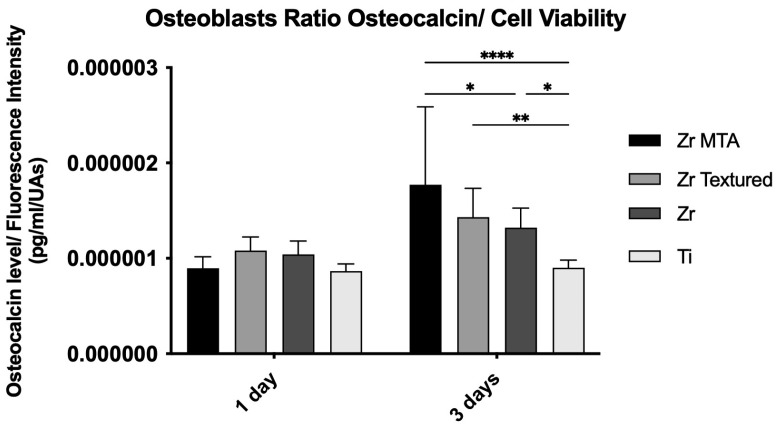
Bar charts showing osteocalcin levels on osteoblasts culture as a mean concentration in pg/mL/UAs. Error bars represent standard deviation. Statistical significance: * *p* < 0.05, ** *p* < 0.01, **** *p* < 0.001; n = 4.

**Figure 7 biomimetics-09-00155-f007:**
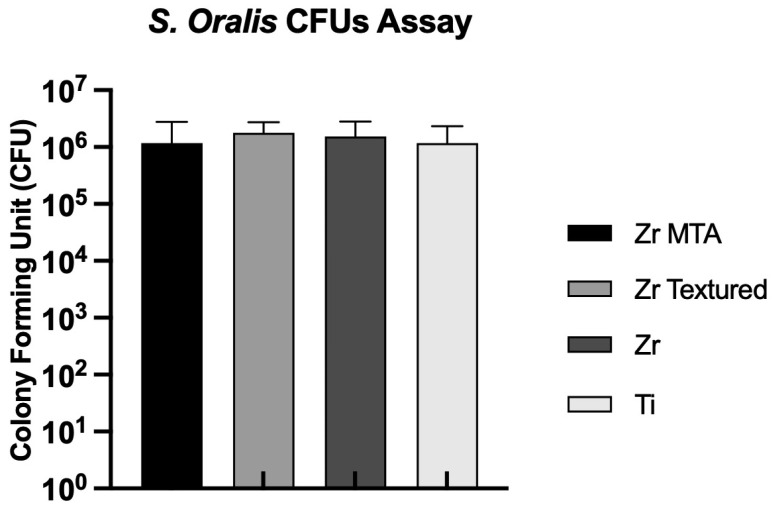
Bar charts showing *Streptococcus oralis* CFUs measured as mean ± standard deviation. Error bars represent standard deviation.

**Table 1 biomimetics-09-00155-t001:** Chemical composition of TZ-3YB-E powder (according to manufacturer Tosoh Corporation©, Tokyo, Japan).

Element	Wt.%
ZrO_2_ + HfO_2_ + Y_2_O_3_	>99.9
Y_2_O_3_	5.15 ± 0.20
Al_2_O_3_	0.25 ± 0.10
SiO_2_	≤0.02
Fe_2_O_3_	≤0.01
Na_2_O	≤0.04

**Table 2 biomimetics-09-00155-t002:** Atomic composition of MTA powder (according to manufacturer MTA Angelus^®^, Angelus, Brazil).

Element	Wt.%
CaO	49.20
SiO_2_	18.58
Bi_2_O_3_	8.26
Al_2_O_3_	4.48
MgO	0.64
SO_3_	0.19
Na_2_O	1.32
Cl	0.51
H_2_O + CO_2_	16.82

**Table 3 biomimetics-09-00155-t003:** Sample groups designation and description.

Samples Designation	Description
Zr MTA	MTA-coated laser-textured zirconia samples
Zr textured	Laser-textured zirconia samples
Zr	Zirconia samples
Ti	Titanium samples

**Table 4 biomimetics-09-00155-t004:** Roughness values—Ra (μm) for Zr MTA, Zr textured, Zr and Ti surfaces as mean and standard deviation.

Sample	Roughness—Ra (μm)	Standard Deviation (μm)
Zr MTA	0.31	0.11
Zr textured	27.73 *	3.22
Zr	0.19	0.14
Ti	0.49	0.14

Statistical significance: * *p* < 0.05.

## Data Availability

Data are contained within the article.

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
