# Peer review of "Cell Biological and Antibacterial Evaluation of a New Approach to Zirconia Implant Surfaces Modified with MTA"

_biomimetics, 2024, doi:10.3390/biomimetics9030155_

Round 1

Reviewer 1 Report

Comments and Suggestions for Authors

The present article proposes a new approach for incorporating MTA into Zirconia implant surfaces with Nd:YAG laser and investigates the biological response of peri-implant cells as well as the microbiological response. 

The article is very interesting.

The topic is very actual.

I suggest the authors to read this review because it is very interesting -

DOI 10.1080/03602532.2020.1758713

Could the authors make figure 3, fig 4 and fig 5  bigger in order to be more lisible.

Could the authors please also mention the limitations of the present study?

In the discussion i would suggest the authors to include more recent published articles.

Comments on the Quality of English Language

Moderate

Reviewer 2 Report

Comments and Suggestions for Authors

This interesting topic of implant surfaces is pertinent. Many nice tests were completed. The authors need to improve the manuscript before it is accepted- please refer to the comments returned.

The authors show they do not understand what MTA products are, but they used one product adequately to make a coating. 

Please improve the overall comparisons of ZrO2 and Ti with less emphasis on the MTA coating. Dentists want to know if ZrO2 is a better implant material.

Comment on why this texture is pertinent and comment on the effects of threads on implants and whether Ti and ZrO2 products have texture like your disks.  

Comments on the Quality of English Language

Many sentences are too long,starting with the title

Many unnecessary words or phrases are included.

Many sentences start with a clause, but would be better if the clause were at the end.

think about keeping it interesting and easy for the reader.

Reviewer 3 Report

Comments and Suggestions for Authors

The authors of the submitted manuscript investigated the response of fetal osteoblasts and immortalized fibroblasts to a laser-structured zirconia surface coated with MTA. Furthermore, the bacterial colonization of this surface was examined. The manuscript provides new information on the cytocompatibility and antibacterial activity of this coating.

However, to improve the manuscript the following points should be addressed:

Title: The term „cellbiological“ would reflect better the topic of the manuscript than “biological”

Abstract: “Zirconia and Titanium discs (Zr and Ti )” à please mention here how these control surfaces were treated (polishing)

Introduction:

Additionally, the collagen fibers orientation of peri-implant connective tissue, 53 mostly circular or oblique, also contributes to the formation of a poor epithelial barrier 54 when compared to Sharpey’s fibers – perpendicular to the tooth surface [3].

è Please clearly differentiate between epithelial and connective tissue attachment, the epithelial attachment is not composed of collagen

2.1.1: For textured surfaces as the laserstructured zirconia Ra strongly depends on the direction of measurement. The measurement of only 3 lines does not seem sufficient. Sa is more appropriate to quantify overall roughness of structured anisotropic surfaces. Therefore, Sa should preferably be calculated according to the currently valid ISO standard.

2.3: Please mention how many replicates of the experiments were performed. Does n=4 mean one experiment with 4 samples?

3.1.1: SEM micrographs of the surfaces (without cells) would be expedient to visualize surface topography.

3.2.1: On osteoblasts culture, Zr MTA samples showed significantly lower viability 274 (p<0.05, one-way ANOVA) unlike Ti group showed significantly higher viability values 275 compared to Zr MTA (1day), to Zr MTA, Zr Textured and Zr (3days) and to Zr MTA and 276 Zr (14days) (p<0.05, one-way ANOVA).

è Error bar at day 14 is missing (Zr MTA – Ti), p-value was calculated via the Tukey´s HSD test and not via ANOVA

3.2.2: Cells are hard to see on SEM images due to the low magnification. Cell morphology and adhesion cannot be assessed on these images. It is thus recommended to insert SEM images with higher magnification.

Figures:

What does an asterisk without bar mean? Please explain in the legend.

In general, for n=4 the error bars appear quite small. Please check if SD was plotted correctly (of if standard error of the mean was plotted instead) and – if necessary – adjust the figure legends.

Figure 5:

Not all statistically significant differences mentioned in the text are marked in the figure. P-value was calculated via Tukey´s test not ANOVA (see above)

Discussion:

Fibroblast differentiation was not investigated, therefore this point should not be discussed.

However, most published studies evaluate the cellular response 357 in contact with MTA cement without being on a textured Zirconia implant surface.

è The coating covers up the zirconia surface. Since the main determinants of the cellular response to biomaterials are topography and (physico) chemistry, these points should be addressed here. It should be explained whether the used surfaces of former studies and the present investigation varied.

The following points should also be addressed in more detail in the discussion:

-        The Zr-MTA surface is smooth. We know that this is not favorable for osseointegration. Why did the authors chose not to create a microrough relief on the surface?

-        IL-8 expression is higher on Zr MTA (osteoblasts). How is this result interpreted by the authors?

Thank you for revising your manuscript!

Comments on the Quality of English Language

Quality of English language requires improvement prior to publication.

Reviewer 4 Report

Comments and Suggestions for Authors

Dear authors,

Thank you for your well-arranged study. Please refer to the following remarks

·         Please include limitations of the study

·         Please compare your results to literature available from the past 5 years

·         Please present more recent literature. Please update references

Round 2

Reviewer 2 Report

Comments and Suggestions for Authors

An interesting set of experiments was submitted. However, the authors did not make the required changes or clarifications.

Be brief; make every sentence count. The first one in the introduction is not needed. They should use "Grammarly" to improve their writing, at a minimum. 

Explain what you mean by "low-temperature aging"- for the audience- which is broad- they may think the "low temp" is below freezing. Did you use this information later- is it pertinent?

Check the punctuation in the abstract. Use the past tense consistently here and after that.

They did not correct the description of MTA. Only some MTA products contain tetracalcium aluminoferrite and bismuth oxide. Why bring up "controversial studies" but rely on the material being antibacterial/fungal? Was this tested for MTA Angelus?

Table 1: It's Y2O3

Explain "green"- it's a broad audience.  

Use "the texturing was performed", not "the texture was performed".

Replace "At the end" with "After sintering, the samples were...".

Table 2: This is the atomic composition and does not agree with the earlier composition description of  MTA given earlier.  You need to clarify for the reader.

Note that the MTA Angelus cement powder is mixed with water and the ratio used. 

What are tube strips? Tubes you cut?

Use "3-4 h", not "about "3-4 h". Use "pressure of the plates", not "grip of the screws". You left the MTA-coated samples in the tubes to ensure setting- not compaction.  

What dental implant brands use Ti6Al4V? Or would Ti have been a better choice for a control? Consistently use "titanium alloy" or Ti6Al4V, not "titanium" or Ti, for your metal samples. Otherwise, you confuse the reader.

Use "measurements were recorded", not "measures were recorded". 

Where is the previous description of the fibroblast cell culture? Add a reference. Use the past tense consistently in this paragraph.

Figure 4 is improved. 

"MTA is extensive" is another unclear phrase; other unclear additions follow in the discussion. Part of the discussion repeats the methods- remove that. MTA samples are not "dried"; they set/harden by reaction with water. Testing the adhesion phrase is unclear...

"MTA into" should be "MTA onto".

Change "does not seem" to "did not show". This would strengthen the impact.  

Comments on the Quality of English Language

Don't use so many capitalized nouns- it's not scientifically appropriate to capitalize elements or compounds or "Dentistry."  Many articles (a, an the) are missing - use Grammarly....

Check the spacing in your manuscript. A spell check should catch that. 

Avoid "there are/is" and "In order to"- both are passive and trite.  

Ofter, the word "different" is superfluous.

Do not resubmit without substantial editing, using an accomplished English writer's assistance.  

Reviewer 3 Report

Comments and Suggestions for Authors

Thank you for revising your manuscript!

Comments on the Quality of English Language

Minor editing is required

Reviewer 4 Report

Comments and Suggestions for Authors

Dear authors,

thank you for your corrections.
